# One Material-Opposite Triboelectrification: Molecular Engineering Regulated Triboelectrification on Silica Surface to Enhance TENG Efficiency

**DOI:** 10.3390/molecules28155662

**Published:** 2023-07-26

**Authors:** Mesude Zeliha Arkan, Zeynep Kinas, Eyup Yalcin, Emre Arkan, Faruk Özel, Abdulkerim Karabiber, Mirosław Chorążewski

**Affiliations:** 1Institute of Chemistry, University of Silesia, Szkolna, 40-006 Katowice, Poland; mesude-zeliha.arkan@us.edu.pl; 2Electrical Engineering Department, Bingol University, Bingol 12000, Türkiye; zeynepkns937@gmail.com (Z.K.); akarabiber@bingol.edu.tr (A.K.); 3Metallurgy and Materials Engineering Department, Ondokuz Mayis University, Samsun 55030, Türkiye; eyyup.yalcin@omu.edu.tr; 4Department of Metallurgy and Materials Engineering, Karamanoglu Mehmetbey University, Karaman 70200, Türkiye; farukozell@gmail.com

**Keywords:** SAM, surface grafting, silica, tunable triboelectrification, TENG

## Abstract

Molecular engineering is a unique methodology to take advantage of the electrochemical characteristics of materials that are used in energy-harvesting devices. Particularly in triboelectric nanogenerator (TENG) studies, molecular grafting on dielectric metal oxide surfaces can be regarded as a feasible way to alter the surface charge density that directly affects the charge potential of triboelectric layers. Herein, we develop a feasible methodology to synthesize organic–inorganic hybrid structures with tunable triboelectric features. Different types of self-assembled monolayers (SAMs) with electron-donating and withdrawing groups have been used to modify metal oxide (MO) surfaces and to modify their charge density on the surface. All the synthetic routes for hybrid material production have been clearly shown and the formation of covalent bonds on the MO’s surface has been confirmed by XPS. The obtained hybrid structures were applied as dopants to distinct polymer matrices with various ratios and fiberization processes were carried out to the prepare opposite triboelectric layers. The formation of the fibers was analyzed by SEM, while their surface morphology and physicochemical features have been measured by AFM and a drop shape analyzer. The triboelectric charge potential of each layer after doping and their contribution to the TENG device’s parameters have been investigated. For each triboelectric layer, the best-performing tribopositive and tribonegative material combination was separately determined and then these opposite layers were used to fabricate TENG with the highest efficiency. A comparison of the device parameters with the reference indicated that the best tribopositive material gave rise to a 40% increase in the output voltage and produced 231 V, whereas the best tribonegative one led to a 33.3% rise in voltage and generated 220 V. In addition, the best device collected ~83% more charge than the reference device and came up with 250 V that corresponds to 51.5% performance enhancement. This approach paved the way by addressing the issue of how molecular engineering can be used to manipulate the triboelectric features of the same materials.

## 1. Introduction

The dramatic increase in the global population and industrial progress of developing countries have resulted in serious problems for the oil and gas sector and have heightened the need for new and green energy sources, which have also become the spotlight of the European Green Deal [1]. One of the most current promising systems is transforming dissipated work to electricity via generators such as triboelectric nanogenerators (TENGs) [2]. These are proposed as useful devices to be applied to both large-scale areas and confined spaces. While having such a broad functionality, the main challenge with this system is the paucity of ideally and industrially adopted materials that can be utilized from miniaturized scientific works to industrial applications [3].

From lab to commodity, the biggest handicap is finding a way for the affordable adaptation of lab methodology to bulk-scale production [4]. This situation requires either setting up a new industrial system for novel lab-based materials or finding a new methodology to enhance the electrical performance of industrially existing materials. The latter is more realistic in terms of processibility and production expenditure. Fascinatingly, the confined properties of existing materials can be extended through polymer grafting by doping, permanent surface processing, and molecular engineering [5,6,7,8]. The combination of the separate features of these approaches can lead to the creation of new durable materials with large energy densities and effective energy conversion that are industrially viable.

Recent developments in TENGs have heightened the need for regulating the dielectric properties of materials that have a great contribution to TENG performance [9]. It has been observed that doping metal-oxides (MOs) to the main structure both improves the dielectric property and reduces the triboelectric losses because the charges formed in the post-contact triboelectric layers are transferred to the additives [10]. This phenomenon reduces the static electron density on the surface and increases the effective gap between the positive and negative triboelectric layers, thereby preventing the fragmentation of the air between the oppositely charged triboelectric surfaces and reducing the recombination of triboelectric electrons [10,11]. A seminal study in this area is the work of Chen et al., in which they doped high-permittivity MOs (SiO_2_, TiO_2_, BaTiO_3_, and SrTiO_3_) as additives in porous PDMS layers to enhance the device output [12,13]. In our previous work, we benefitted from the advantages of shape-controlled MOs to compare the structure-driven electrical features of MOs [9]. In our previous follow-up study, we showed how the molecular structures manipulated the TENG parameters [5]. Despite all of this, all these distinct studies are still limited in terms of a synergistic evaluation of the advantages of these separate approaches. Therefore, the benefits of molecular engineering to enhance the existing features of MOs can be proposed as a convenient method.

According to the Lewis acid–base theory, molecules with unpaired electrons, such as oxygen and sp^3^ hybridized nitrogen, have high electron-donating properties and are mainly regarded as tribopositive enhancers [14,15]. Specifically, their application as self-assembled monolayers (SAMs) can improve the charge density of the attached surface by providing an external dipole moment, thereby changing the surface electrochemical properties, such as the work function and surface potential [16,17]. The best examples would be triphenylamine and carbazole-based SAMs, which are widely used in organic electronics due to their high load-carrying capabilities [18,19]. These types of phenyl amines donate electrons to metal-oxide surfaces, but also regulate their hole-carrying ability, as their nitrogen atoms can be easily oxidized [20,21]. Moreover, SAMs also have the advantage of easily adjustable electronic structures to contribute load flow by changing the fermi level of the surface on which they are chemically bonded [18,22,23]. Therefore, they are mainly preferred in the synthesis of high-performance hybrid structures. A properly designed SAM to modify inorganic materials results in the formation of an ideal hybrid structure with the desired electron-donating and -withdrawing ability [24]. This type of hybrid material can be used as a constructive additive to the main matrix to improve the polarized charge by providing electrons or holes to the triboelectric structure, enhancing the triboelectric charge between the two layers, and helping to improve the output voltage. Nevertheless, while these benefits make SAMs ideal, their utilization has yet to come into prominence in TENG fabrication.

In the present work, therefore, we focused on the longitudinal evaluation of the combined features of both organic and inorganic materials by proposing new hybrid structures. Our hypothesis is based on the surface modification of MOs to fine-tune their triboelectric properties that can be used as dopants in different polymer matrices to enhance their charge potentials and to monitor their performance in device operation. Therefore, we have specifically chosen silica (SiO_2_) as the target MO because its utilization in research, such as in energy studies, and in the industry is widespread [25,26,27]. The strategy for the grafting of SiO_2_ is regarded as model work that can be accepted to be applied to any other MOs to tune its electrical properties. To this end, various phenylamine- and fluorosilane-based SAMs have been chosen to load different electron-donating and -withdrawing abilities to target surfaces. Although the optimum grafting density for precipitated silica has previously been reported as a minimum of 4 µmol/m^2^ for silanes, we have decided to use 5 µmol/m^2^ to be sure of sufficient surface modification because we have different surface-attaching groups for distinct SAMs [28]. Additionally, nylon 6.6 and PAN fibers have been formed for each triboelectric layer to enhance their possible applicability to different areas. The effect of various grafting on the surface morphology, wettability, and electrical performance in TENG devices has been discussed. The present work fills the research gap in the literature by indicating molecular engineering-regulated tunable triboelectrification that can be applied from the lab to bulk-scale materials processing.

## 2. Results and Discussions

Figure 1 is an example of the schematic indication of the sol-gel synthesis of hybrid nanoparticles, their preliminary mixture with the matrix, and the electrospinning process of the triboelectric layers, respectively. As the opposite dielectric layers, PAN and nylon 6.6 were intentionally selected due to their superior benefits, such as enhanced volume, cost efficacy, small pore size enabling high porosity, and large specific surface area [5,29,30].

X-ray photoelectron spectroscopy (XPS) was used to determine the elemental composition and atomic bond formation on the SiO_2_ surfaces. Appendix A shows the survey spectrum of ME-86, ME-103, ME-111, and bare SiO_2_. All the elements of each sample have been successfully obtained and shown in the figures. To confirm the atomic bonds formed by the carbon and oxygen atoms on the SiO_2_ nanoparticles after molecular grafting, a high-resolution surface spectrum (HRSS) of C1s and O1s was analyzed. Figure 2 shows the HRSS of C1s and O1s for ME-86, ME-103, and ME-111. The C1s spectra were decomposed into three peaks corresponding to carbon in different chemical environments. The binding energy peaks of C1s were assigned to C–C/C–H at 284.6 eV (284.4 eV and 284.5 eV), C–O–C at 285.9 eV (285.9 eV and 286.2 eV), and O–C=O at 287.4 eV (287.9 eV and 288.1 eV) for ME-86, ME-103, and ME-111 (Figure 2 up row). In addition, the C1s of ME-86 have CF_2_ and CF_3_ at 291.5 eV and 293.7 eV, respectively, due to its fluorocarbon-based molecular structure [19,31].

The O1s high-resolution survey spectrum of all the samples is shown in Figure 2 (bottom row). A strong signal centered around 533 eV is due to the silica substrates for all the samples and represents the lattice oxygen of each oxide [32]. The peaks at 531.2 eV and 531.6 eV represent the C=O (carbonyl group) from the oxygen of the organic molecules on ME-103 and ME-111 and unavoidable adventitious carbon on the substrates, respectively (all the corresponding bonding energy of the functional groups is presented in Table 1).

The XPS measurements were also carried out on bare SiO_2_ prior to the modification with molecules. The HRSS of C1s and O1s for bare SiO_2_ are shown in Appendix A. For C1s, the same functional groups and corresponding peaks were obtained almost at the same signal values as the signals of ME-86, ME-103, and ME-111. These carbon contributions are due to unavoidable adventitious carbon (surface contamination) [33].

Figure 3 shows the HRSS of N1s for ME-86, ME-103, and ME-111. The binding energy of the N1s is around 400 eV. As there is no nitrogen in ME-86 and bare SiO_2_ substrate, no N1s signal was recorded. Therefore, the absence of the N1s peak for ME-86 and N1s peaks’ presence on the ME-103 and ME-111 confirms that the SiO_2_ nanoparticles successfully modified with the molecules. The FT-IR spectra of ME-103 and ME-111 are available in Appendix A.

In particular, ME-86 includes ‘fluorine’ and ME-103, and the ME-111 samples include ‘boron’ atoms. Appendix A shows the HRSS of F1s at 688.5 eV for ME-86 and B1s at 193.3 eV and 193.5 eV for ME-103 and ME-111, respectively. These specific peaks again confirm that the proposed organic structures have been successfully attached to the surface of the SiO_2_ particles. In addition, the HRSS of Si2p is shown in Appendix A. The peaks of Si2p for ME-86 at 104.8 eV, for ME-103 at 105.2 eV, for ME-111 at 104.3 eV, and for bare SiO_2_ at 104.3 eV are attributed to Si–O; all these peaks are signals of the SiO_2_ substrates. Furthermore, the O-Si peak at 102 eV is a signal of the Si of the ME-86 molecule and the O of the SiO_2_ substrate (Appendix A).

To compare the difference between the fabricated nanofibers, an SEM analysis of the fiber structures has been carried out, and the formation of the film quality on the electrode has been evaluated. Figure 4 shows the morphology of the undoped, bare silica, and ME-103- ME-111-doped nanofibers. As depicted in the SEM images, all the undoped nanofibers have a uniform distribution with smooth surfaces, whereas the heterogeneity in the fiber diameter and film surface increase after bare silica doping. This is specifically apparent in the case of 1% doping, where the fluctuation in the fiber size is higher with respect to 2% and 3% due to possible uneven perturbation in the surface charge in the polymer solution, which also leads to axisymmetric and non-axisymmetric variability and uneven stretching of the jet in whipping [5,34,35]. Fortunately, this problem was alleviated by doping ME-103 and ME-111, which reduced the fiber diameter and enhanced the homogeneity throughout the film surface. The possible explanation for this is that surface-modified silica nanoparticles show interface activity and decrease the surface tension of the mixture. Moreover, the fiber diameters and surface tension are directly proportional with each other in accordance with the mathematical model [5,36,37]. Exceptionally, 3% doping of ME-111 follows another trend in which thicker but still homogeneous fibers formed. This could be due to the late coagulation of the polymers leading to the adhesiveness of the fibers and evaporation of the solvent after the deposition of the fibers on the collector [38]. Concerning the SEM images of the ME-86- and ME-88-doped fibers, the homogeneity and size of the fiber structures were maintained in all the dopant ratios. It has been seen here that the dopant ratio merely regulated the formation of fibers with particular textures. However, local agglomeration and detrimental cavities are more prominent after 3% doping in ME-86 and 2% doping in ME-88 (for details, see Appendix A).

The surface morphologies and contact angles of the fibers are essential factors to obtain further insight into the surface characteristics of the produced films and the TENG outputs associated with them [39,40,41]. Figure 5 and Figure 6 represent the AFM images and the measured contact angle values of the dielectric films and Table 2 summarizes the root-mean-square (Sq) roughness and surface energy values of the fiber films. It is worth noting that the optimized roughness with the desired surface morphology is constructive for determining the efficient friction force through increased roughness on the surface that leads to an enhanced charge transfer [41,42,43]. Otherwise, the formation of air traps and destructive cavities is a handicap for an effective contact surface area. In addition, the surface energy mainly regulates the contact angle and the surface roughness contributes to the contact angle values [28]. In general, a higher surface energy corresponds to higher wettability on the surface, as well as an increase in the roughness, which makes a hydrophobic surface even more hydrophobic and a hydrophilic surface even more hydrophilic [44,45,46,47]. The framework of this information clearly explains why the doping of bare silica resulted in a slight decrease in the contact angle values since its addition caused an increase in the roughness of hydrophilic nylon and, therefore, increased wettability. More fascinatingly, knowledge in the literature has proven that donor semiconductors using boronic acid as an anchoring group create a dipole moment to the metal-oxide surface and enhance its electron density [48]. In comparison to bare silica, this circumstance creates more electron density on the particle surfaces. The increase in the roughness together with electron density throughout the nylon matrix also led to an even greater increase in the surface energy and wettability. Therefore, it can be deduced that alterations in the wettability values may be separately related to the surface topography and changes in the interfacial charge characteristics of the created fibers (see Appendix A for the surface characteristics of the ME-86- and ME-88-doped fibers).

### TENG Performance

In this study, silica nanoparticles were subjected to surface grafting for the fine-tuning of the triboelectric behavior of the particle surface to obtain task-specific materials. In general, metal oxide surfaces are rich in several active moieties or dangling bonds, such as hydroxyl, hydrogen, and carbonyl groups, which cause the creation of surface trap states, thereby decreasing the charge recombination velocity after the charges are polarized during device operation [23,49,50,51]. In addition, the polymer films of the triboelectric layer can possess some unsaturated sides that can bond chemically with the metal oxide surface and can cause trapping and de-trapping of the carriers by changing the electron density in the polymer film [52,53,54]. Beyond the charge-trapping behavior, we hypothesized that the performance of the organic semiconductor modification is already proven as a versatile tool to improve the electron density of anode and cathode surfaces in photovoltaic studies [55,56,57]. Similarly, this approach can be utilized to enhance the electron density and electron transfer ability of dielectric nanoparticles that are used as charge trappers in the tribopositive layer. To this end, CRB and H-TPA structures were specifically chosen as surface modifiers since they have been frequently used as strong electron-donating molecules in the literature [21,58]. This approach enabled us to combine the high permeability of metal oxides with electron-donating structures that significantly improved the charge extraction ability and enhanced the device parameters, such as the current, voltage, and power. Additionally, PAN, nylon 6.6, and silica were specifically chosen as the main materials because their production and utilization are already adopted by the research and industry and cover a broad spectrum of applications in the market [59,60,61,62,63,64,65]. We postulated that enhancing the triboelectric properties of such polymers rather than producing new and expensive ones and indicating both the synergistic and antagonistic sides of the method could be a model work to develop a novel strategy that can be applied to any other technically accepted materials.

The most well-known phenomena explaining electron movement and charge accumulation within the molecular structure are resonance and inductive effects [66,67,68,69,70]. The resonance effect is regulated by conjugated bonds and results in the direct movement of electrons from one side to another, while the inductive effect is driven by electronegativity differences within the bond and gives rise to different electron densities in pairs of atoms bound by the same bond [66,71,72]. Herein, the H-TPA molecule utilizes the benefits of these two effects, as it has tri-phenyl conjugation groups supported by inductive positive (+I) hexyloxy chains [5,21,66,73]. As for CRB, it only has the conjugated phenyl rings, whereby it executes the electron transfer. In accordance with this knowledge, H-TPA is expected to be more favorable in terms of the electron donor ability than CRB. This issue has been proven in early research and has already been confirmed by our recent work calculating the HOMO level from the oxidation potential through cyclic voltammetry analysis, in which bare TPA showed a lower oxidation potential than bare CRB; it was shown that even +I methoxy bearing TPA has a lower oxidation potential than bare TPA on an indium tin oxide surface [18,21]. In the present case, both the ME-103 and ME-111 structures were obtained by chemical grafting of the H-TPA and CRB groups to SiO_2_, respectively. These novel hybrid structures have high oxygen concentration and sp^3^ hybridized nitrogen behaving as a Lewis base; therefore, they are anticipated to improve the electron density on the SiO_2_ surface that also create the electron-trapping region in the nylon 6.6, triggering better polarization and passivating the charge recombination [15]. As far as ME-86 and ME-88 are concerned, they merely possess the advantage of the inductive negative (−I) nature of the fluoroalkyl chain to regulate the electron density on the surface. It is worth noting that NMR studies in the literature state that the inductive effect becomes weaker from the C1 side to the Cn side as the carbon chains become longer and is finally regarded as ignorable, specifically after the third carbon chain [5,66,74,75]. Therefore, the triboelectric behavior of the ME-86- and ME-88-doped polymers can be expected to be regulated by the remaining factors, such as an effective content area, the quality of the obtained nanofibers, and the formation of the tight packaging of the molecules on the surface, because previous studies indicate that as the number of Si_(silica)_-O-Si bonds on the silica surface increases, the tribonegative properties increase, as well [76].

Figure 7a–c indicates the instant output voltage of the TENGs fabricated using a series of doped nylon and undoped PAN. Herein, the device made up of pure nylon 6.6 and pure PAN is accepted as a reference device. The maximum output voltage of the reference device was measured as 165 V, while the voltage value after doping of the various amounts of (1, 2, and 3%) bare silica has been measured as 208, 207, and 150 V, respectively. It was seen that the doping of bare silica caused improvement in the device voltage until a certain amount of dopant ratio. Moreover, the doping of ME-103 and ME-111 gave rise to further recovery in the output voltage, and the maximum values were measured as 231, 220, and 122.9 V for 1, 2, and 3% ME-103 doping, whilst these values have been found as 212, 221, and 190 V for 1, 2, and 3% ME-111 doping. In the trials, ME-103 and ME-111 tended to have greater performance improvement with less amount of doping than bare silica; this improvement was dominant in the case of ME-111 doping. We can infer from these data that the strongest donor ability of ME-103 has driven the best tribopositive nature to the polymer matrix even with the least amount of doping.

The charging behavior of the TENGs was analyzed for further evaluation of the device outputs by connecting the TENG to the 0.3 µF capacitor subtending to a single pulse; the obtained results are indicated in Figure 7d–f. For the same period of time (16 s), the measured capacitive voltage of the reference device was found at 6.55 V, whilst this value increased to 7.59, 11.90, and 10.4 V for 1% doped bare silica, 1% doped ME-103, and 2% doped ME 111, respectively. By means of the equation of Q = CV, the stored amount of charge was calculated for the same devices, in turn, as 1.97, 2.28, 3.57, and 3.12 nC. The measured instant output voltage and the calculated highest capacitive voltages are consistent with each other because one of the important factors affecting the output performance of TENGs is the matching of impedance between the load and the fabricated device. Basically, the output performance increases under operating conditions when the capacitive and ohmic properties of the load impedance are close to the impedance properties of TENG. However, there is a non-linear relationship between the load impedance and the output voltage; the change in material content or surface area of the applied materials can affect the output voltages of the ohmic and capacitive loads differently, as well. Namely, the output voltage of each or only one of the ohmic and capacitive loads may increase or decrease depending on the operating conditions and material content. In accordance with this information, in the present study, the highest voltages in the ohmic and capacitive loads were obtained with the same doping ratio, while the lowest output voltages were obtained with various doping ratios. In addition, the effect of the doping ratios on the output voltage at the ohmic and capacitive loads were found to be varied.

The correlational analysis of the current–voltage curves under varying loads is depicted in Figure 8a–c. It is clear from the graphs that the output current was restricted by an increase in the applied loads due to the limitation of the current by the ohmic losses, and a dramatic reduction appeared until the applied load increased to 4.7 MΩ. In addition, since there was a slight effect of the load up to the saturation point (SP), the voltage progressively increased up to this point [9]. Passing over the SP, owing to a further increase in the load, inversely influences the current; this is why the voltage starts to decrease. Once the applied load reached 66.4 MΩ, the reference device showed the maximum value of 398.4 V. Fascinatingly, 1% doping of the bare silica led to a maximum output of 431.6V, while ME-103 and ME-111 produced the maximum voltages of 524.56 and 484.72 V for 1 and 2% doping, respectively. In the same way, doping caused a dramatic increase in the current values. The reference device showed a maximum of 27.3 µA under 1.1 MΩ. After the doping process, the current values were subjected to a ~22, 70, and 55% increase and measured, in turn, as 33.3, 46.4, and 42.4 µA for bare silica, ME-103, and ME-111. In compliance with the literature evidence and outcomes of the methodology in the present study, it can be stated that the proposed hybrid structures are not only beneficial in terms of preventing charge recombination, but are also favorable for providing essential surface charge density to enhance the electron transfer ability. This is specifically true in the case of ME-103, which has a high electron-donating ability with respect to ME-111 and bare silica. This could be the reason why ME-111 provided the best electrical values with the minimum amount of doping.

The output power of the fabricated devices has provided deeper insight into the contribution of the dopant to device performance. The produced power values were calculated through the equation of P = V^2^/R depending on the changing loads, and the obtained data set is indicated in Figure 8d–f. The best power values were found when the load was 19.4 MΩ for the reference and the other devices made up of 1% doped ME-103 and 2% doped ME-111 that produced 3.8, 5.7, and 5.0 mW, respectively. However, bare silica reached its highest power value of 4.3 mW at 10 MΩ for 1% doping. Among the hybrid samples, ME-103 had the highest power density at 3.6 W/m^2^ (Figure 9). All the power curves follow similar trends with altering loads and approach 0 after their maximum points. These results suggest that the doping of hybrid particles has non-negligible contributions to the charge density of the materials and enhances the charge generation, improving the electrical power (see Appendix A for device parameters of the ME-86- and ME-88-doped fibers).

Taken together, the best-performing combination of tribopositive and tribonegative materials has been chosen to fabricate a new device with the best enhancement in electrical parameters with respect to reference one, and the obtained results are given in Figure 10. The instant output voltage increased by 51.5% and was obtained as 250 V. Herein, the capacitive voltage was measured with various capacitors to find the optimum relation between the stored charge and the capacitive voltage. The capacitive voltages were found, in turn, to be 12, 9.02, 8.73, and 7.5V for 0.3, 0.5, 0.7, and 1 µF for the same period of time. The stored charges for the same capacitors were calculated as 3.6, 4.5, 6.11, and 7.5 nC, respectively. In the trial, the best device collected ~83% more charge than the reference. Additionally, as in the aforementioned cases, the best current and voltage were attained at 1.1 and 66.4 MΩ and measured as 69.7µA and 650.7 V, respectively. The best power and power density were calculated as 8.80 nW and 5.50W/m^2^ at 19.4 MΩ. Finally, the device was subjected to 20 N at 4Hz to evaluate whether there is a notable change in the instant voltage. The voltage was measured at 379 V with the best device combination. Consequently, the non-negligible increase in the electrical outcomes is associated with the tuning of the interfacial charge properties of the polymer-based fiber (see Video S1 for the practical application of the best-performing device.)

## 3. Materials and Methods

### 3.1. Materials

(4-(9*H*-carbazol-9-yl)phenyl)boronic acid (CRB), nylon 6.6, polyacrylonitrile (PAN), *N*,*N*-Dimethylformamide (DMF), chloroform, formic acid, and acetonitrile were obtained from Sigma-Aldrich (Burlington, MA, USA). SiO_2_ nanopowder (10–20 nm (TEM), 99.5% trace metals basis) was purchased from Sigma-Aldrich. The procedure for the synthesis and purification of (4-(bis(4-(hexyloxy)phenyl)amino)phenyl)boronic acid (H-TPA) was previously introduced to the literature by our research group [77]. Perfluorodecyl-1*H*,1*H*,2*H*,2*H*-dimethylchlorosilane (PFDec-MCS) and Nonafluorohexyldimethylchlorosilane (FHex-MCS) were obtained from Gelest Inc. (Morrisville, NC, USA).

### 3.2. Synthesis of Hybrid Nanoparticles

Approximately 250 mg of two silica samples (as received) were suspended in 20 mL of chloroform in two separate 250 mL round-bottom flasks. Then, 299.6 mg of H-TPA and 198.5mg of CRB were dispersed in 5 mL chloroform, assuming to graft 5 µmol/m^2^, and added dropwise to the flasks while the solutions were stirred and refluxed overnight. After the silica particles were recovered by filtering at room temperature under a vacuum, they were purified by extraction thrice using equal volumes of pure hexane and pure dichloromethane, respectively, to ensure the elimination of any physically bound organic moieties and other surface impurities. Finally, the silica particles were collected, placed in vials, and dried at room temperature for 1 day. A similar synthetic route and purification method was used for the fluorocarbon modification of the silica surface. However, the only difference was that the amount of the silane modifiers was selected to be 452 µL and 176 µL for PFDec-MCS and FHex-MCS, respectively, to arrange the surface grafting density of 5 µmol/m^2^. A schematic illustration of the reaction mechanisms for each synthetic step is indicated in Appendix A.

### 3.3. Electrospinning Process

The tribopositive layers were produced by dissolving four separate (1 g) nylon 6.6 specimens in different chloroform/formic acid mixtures (5 mL/5 mL) and adding various amounts of (0, 1, 2, and 3 wt%) bare silica, ME-103, and ME-111 to distinct nylon solutions. Each final mixture was stirred overnight until a homogeneous solution was formed, and the proper viscosity was obtained via electrospinning. Then, the final solutions were transferred to the syringe by using a 21-gauge metallic needle tip. A power supply with a direct current (DC) source was used to carry out the electrospinning process. The distance between the collector and needle tip was set to 15 cm, and the voltage and solution feeding rate were set as 18 kV and 0.1 mL/h, respectively. Finally, the produced nanofibers were collected on aluminum foil.

As for the tribonegative layers, 3 different (1 g) PAN samples (10% *w*/*v*) were dissolved in 10 mL DMF and, subsequently, distinct amounts of (0, 1, 2, 3, and 4 wt%) ME-86 and ME-88 were added to the beaker and homogenized by stirring for 24 h. The formation of the fibers was implemented by following the same procedure (but at a different feeding rate of 18 kV and 1 mL/h) for the fiberization of the nylon 6.6 polymers.

### 3.4. Device Fabrication

The supporting layers of TENG are made up of plexiglass with dimensions of 5 (H) × 100 (L) × 100 (W) mm^3^. Doped nylon 6.6 and PAN fibers were used, in turn, as positive and negative dielectrics. Each triboelectric layer was cut into the dimensions of 40 (L) × 40 (W) mm^2^ and stuck on the plexiglass layers. Each corner of TENGs was supported by a mini spring and they were fixed by drilling holes and gluing them in the bottom and upper sides of the layer. The electrical output of the TENGs was measured by a hydraulic pressing machine equipped with a computer and software system applying a constant force of 20 N and a frequency of 2 Hz. The equivalent resistance was obtained by considering the internal resistance of the oscilloscope (10 MΩ) and used in the power calculations of the fabricated devices. In the electrical measurements, the AC waveform of the devices was converted to a DC waveform through a bridge diode enabling us to charge the capacitances. The working principle of the device is shown in Figure 11.

### 3.5. Characterization

A Thermo Scientific Model K-Alpha XPS (Thermo Fisher Scientific, Waltham, MA, USA) instrument was used to conduct the XPS tests using monochromatic Al K radiation. High-resolution survey spectrum scans for each element were obtained with a pass energy of 50 eV with a 0.1 eV energy step size, and the survey spectra scans were obtained with a pass energy 200 eV with a 1 eV step size. In addition, the Thermo Scientific Nicolet Summit (Waltham, MA, USA) Fourier transform infrared spectrometer (FTIR) was used to confirm the chemical bonding on the surface. An analysis of the nanofiber structures was carried out by Zeiss Sigma 300 VP (Oberkochen, Baden-Württemberg, Germany) scanning electron microscopy (SEM) using the following parameters: magnification = 19.79–20.21 KX, EHT = 5.00 kV, WD = 4.7–7.2 mm, resolution = 1 μm. The wettability of the film surfaces was measured with a Biolin Scientific Theta Lite optical tensiometer (Stockholm, Sweden). The surface topography of the samples was measured with a Nanosurf Easyscan 2 Controller (Liestal, Switzerland) Atomic Force Microscopy (AFM) with non-contact mode and a scan speed (time/line) of 1 μm/s to obtain a 256 × 256-pixel image. All the measurements were carried out at room temperature using cantilevers with the following nominal properties: thickness = 7 μm, length = 225 μm, width = 38 μm, resonance frequency = 190 kHz, and force constant = 48 N/m. The electrical parameters of the fabricated TENG devices were recorded with a Rigol MSO5104 digital oscilloscope (Rigol Technologies, Beijing, China).

## 4. Conclusions

In this study, we showed a feasible molecular engineering method to tune with triboelectrification on particle surfaces and to enhance the charge potential of triboelectric layers and related TENG parameters. Since silica is commonly used for both research and industry due to its convenient chemical stability, high relative permeability, and, especially, charge-trapping ability, its surface modification is preferred as model work to tailor surface triboelectric properties, which then can be applied to any other metal oxide. Likewise, nylon 6.6 and PAN were chosen as opposite triboelectric polymer matrices for impregnation and fiberization to enhance their applicability to a wide range of areas. Each hybrid structure has a unique physicochemical characteristic; therefore, its doping exhibited independent surface topography, roughness, and wettability. Regardless, the best dopant and their ideal dopant ratio have been studied for each triboelectric layer. The results indicate that 1% doping of ME-103 provided with the best electrical performance and better electrical output than even 2% doping of the ME-111 counterpart due to the superior electron-donating feature of the H-TPA over CRB, while ME-88 showed a better electrical performance than ME-86, possibly owing to its better surface coverage and Si_(Silica)_-O-Si formation ability that enhanced the tribonegative nature of the silica. Compared with the reference device, separate doping of (1%) ME-103 to a tribopositive layer and (2%) ME-88 to a tribonegative layer led to ~30.2% and ~33.3% increase in the instant voltages and ~70 and ~48% increase in the current values, respectively. Likewise, their doping gave rise to a ~50 and ~34.2% rise in the obtained powers. Among all the fabricated TENGs, the best device parameters were measured by the combination of the best triboelectric layers. For the champion device, a ~51.5% increase in the voltage and an ~83% higher charge collection ability were measured with respect to the reference device. It was also observed that when the dissipated work increased to 4 Hz (20 N), the system noticeably enhanced the electrical parameters and led to an instant measurement of 379 V.

It has been found that changing the triboelectric behavior of silica by surface grafting with proper self-assembled monolayers is feasible in terms of practical applications. Together with this knowledge, several factors, such as temperature, humidity, application force, dielectric constant of the material, and effective surface area, collectively affect the output performance of TENGs. Although constant environmental conditions were maintained throughout the experiments, the main weakness of this study is the paucity of analysis indicating the change in the dielectric properties and effective surface areas of the opposite layers before and after doping. Therefore, more research must be conducted to overcome these limitations.

Our research related to molecular engineering-guided TENG fabrication is ongoing in our laboratories. To be more specific, further studies regarding the role of the hole- and electron-transport materials in triboelectric layers would be worthwhile. We are currently evaluating their effect on TENG device performance. We envision a broader and more detailed identification of the molecular-tailored electrical output; however, detailed molecular mechanisms and their optimum device parameters will be the subject of our upcoming work.

## Figures and Tables

**Figure 1 molecules-28-05662-f001:**
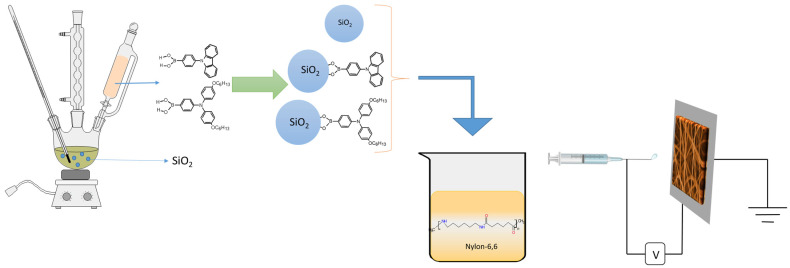
Schematic illustration of hybrid material synthesis and fiberization process.

**Figure 2 molecules-28-05662-f002:**
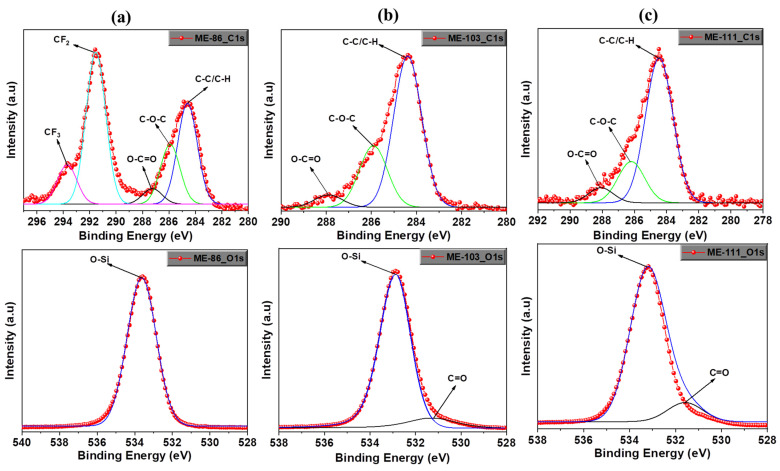
XPS high-resolution surface spectrum of C1s (up) and O1s (down) for ME-86 (**a**), ME-103 (**b**), and ME-111 (**c**).

**Figure 3 molecules-28-05662-f003:**
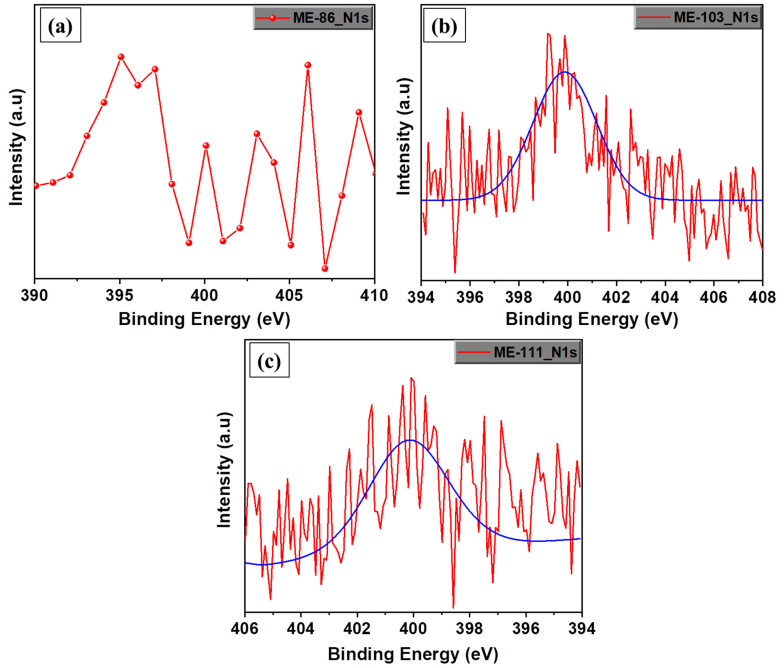
XPS high-resolution surface spectrum of N1s for SiO_2_/ME-86 (**a**), SiO_2_/ME-103 (**b**), and SiO_2_/ME-111 (**c**).

**Figure 4 molecules-28-05662-f004:**
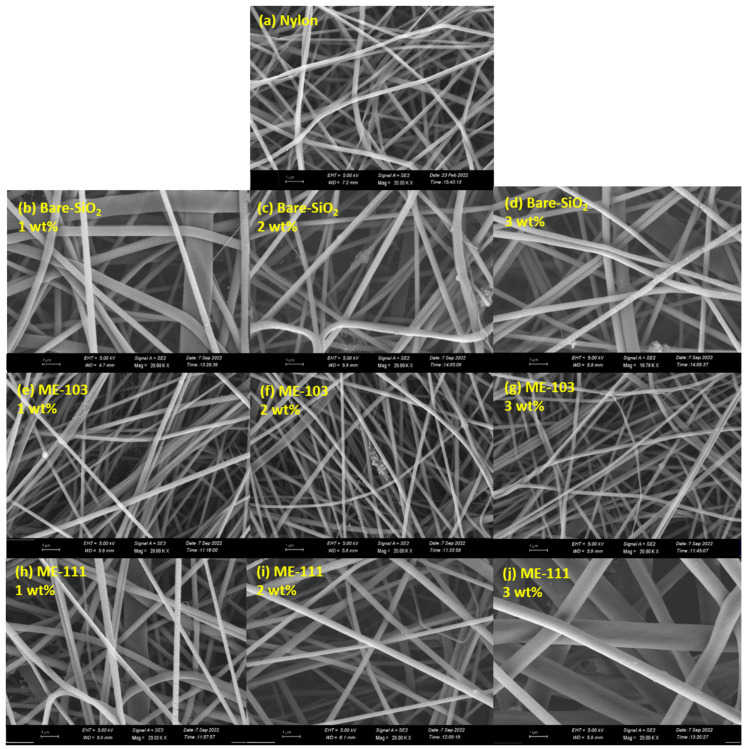
SEM images of tribopositive fibers: nylon (**a**), 1 wt% bare SiO_2_ (**b**), 2 wt% bare SiO_2_ (**c**), 3 wt% bare SiO_2_ (**d**), 1 wt% ME-103 (**e**), 2 wt% ME-103 (**f**), 3 wt% ME-103 (**g**), 1 wt% ME-111 (**h**), 2 wt% ME-111 (**i**), 3 wt% ME-111 (**j**).

**Figure 5 molecules-28-05662-f005:**
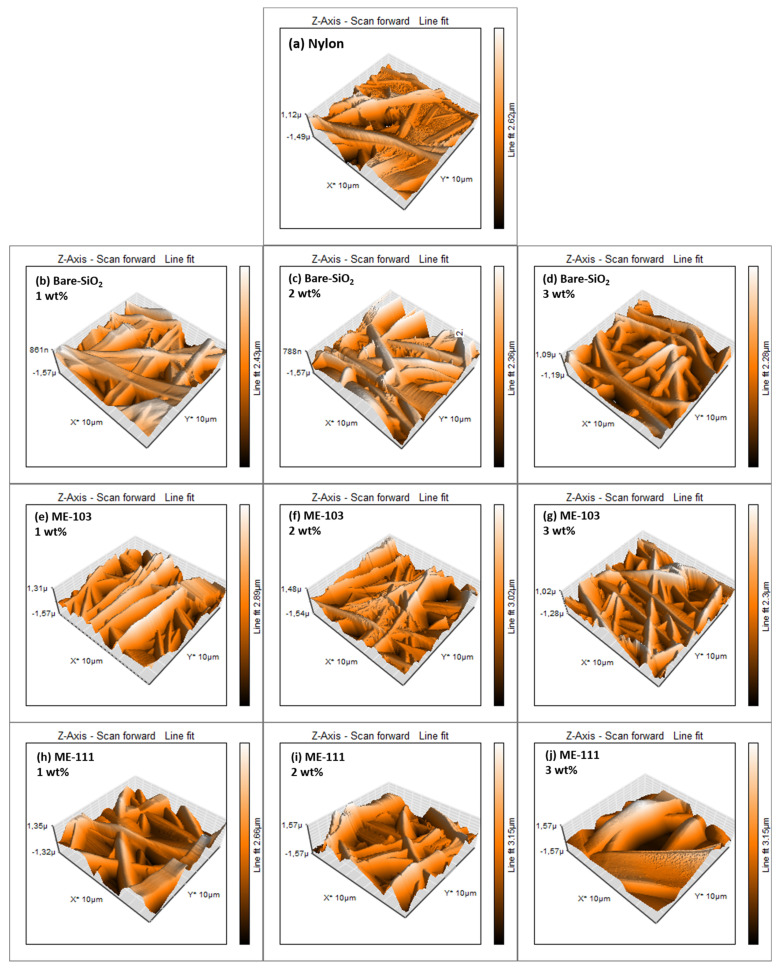
Topographical AFM images of tribopositive fibers: nylon (**a**), 1 wt% bare SiO_2_ (**b**), 2 wt% bare SiO_2_ (**c**), 3 wt% bare SiO_2_ (**d**), 1 wt% ME-103 (**e**), 2 wt% ME-103 (**f**), 3 wt% ME-103 (**g**), 1 wt% ME-111 (**h**), 2 wt% ME-111 (**i**), 3 wt% ME-111 (**j**).

**Figure 6 molecules-28-05662-f006:**
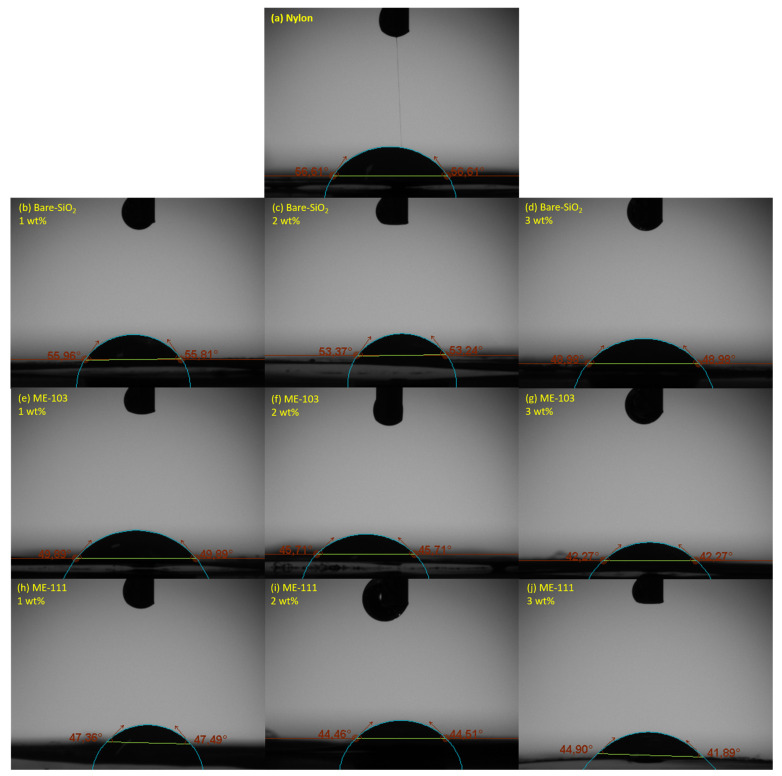
Contact angle images of tribopositive layers: nylon (**a**), 1 wt% bare SiO_2_ (**b**), 2 wt% bare SiO_2_ (**c**), 3 wt% bare SiO_2_ (**d**), 1 wt% ME-103 (**e**), 2 wt% ME-103 (**f**), 3 wt% ME-103 (**g**), 1 wt% ME-111 (**h**), 2 wt% ME-111 (**i**), 3 wt% ME-111 (**j**).

**Figure 7 molecules-28-05662-f007:**
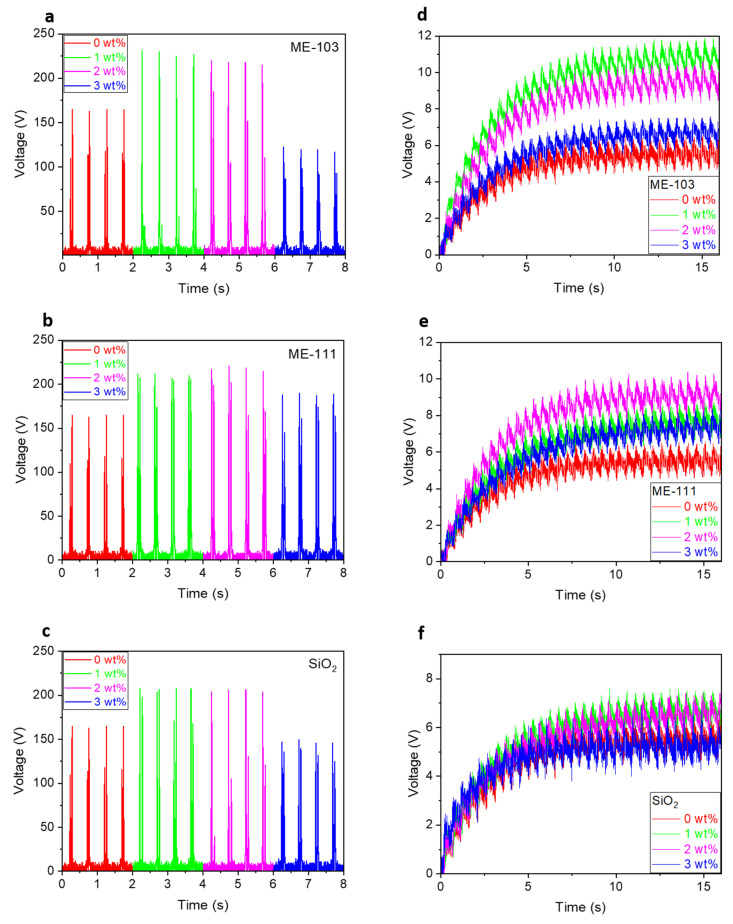
Instant output voltages under 20 N force at 2 Hz frequency (**a**–**c**) and capacitive voltages (**d**–**f**) of bare SiO_2_ and ME-103- and ME-111-doped TENG devices.

**Figure 8 molecules-28-05662-f008:**
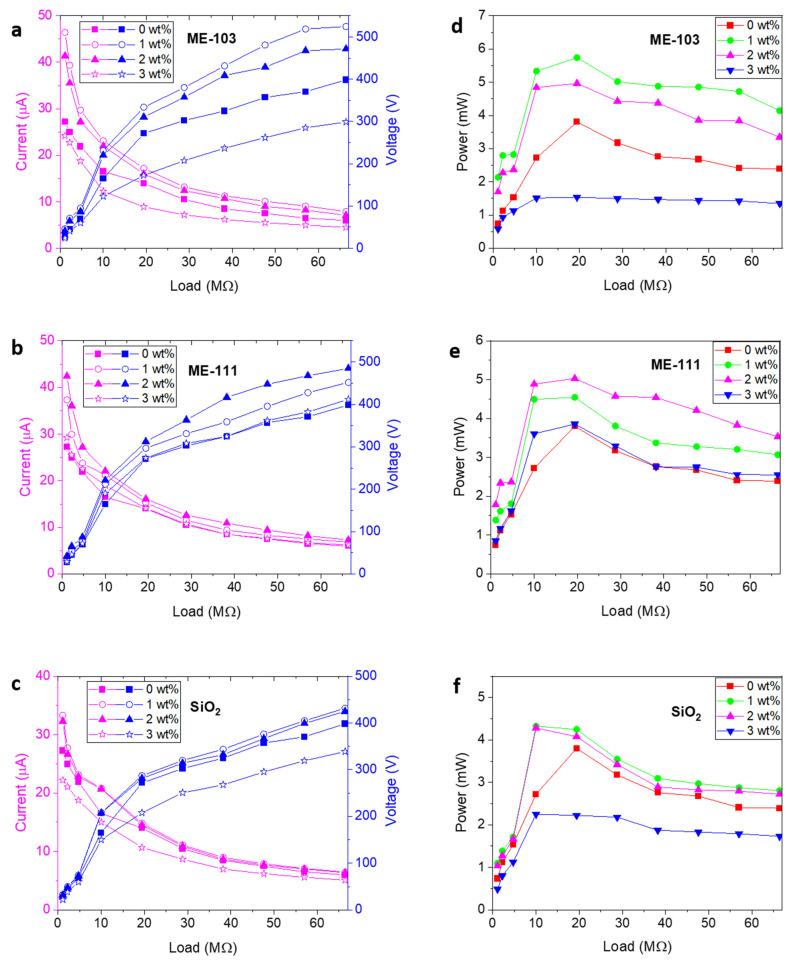
Current–voltage (**a**–**c**) and power (**d**–**f**) curves of (under 20 N force at 2 Hz frequency) bare SiO_2_ and ME-103- and ME-111-doped TENG devices under different loads.

**Figure 9 molecules-28-05662-f009:**
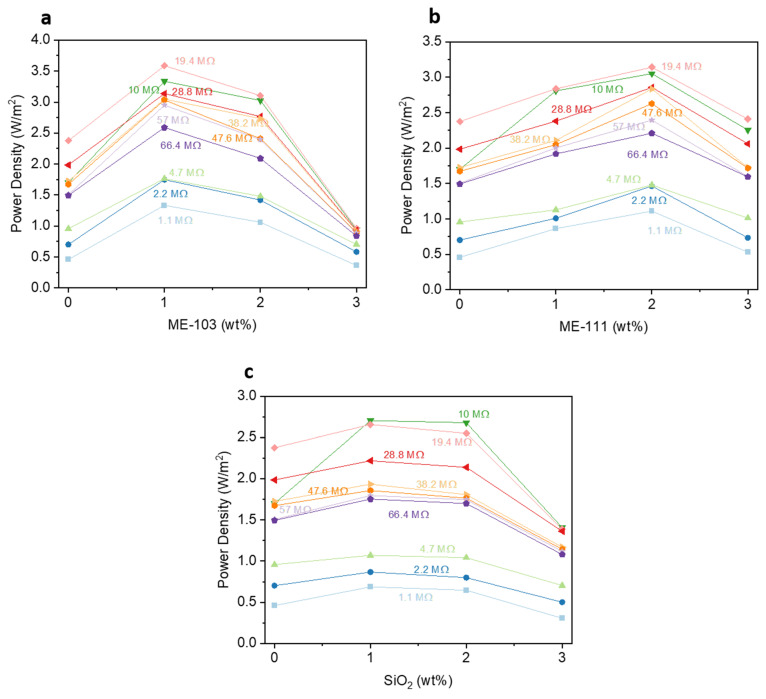
Power density curves of ME-103 (**a**), ME-111 (**b**), and bare SiO_2_ (**c**) doped TENG Devices for varying Doping Ratios.

**Figure 10 molecules-28-05662-f010:**
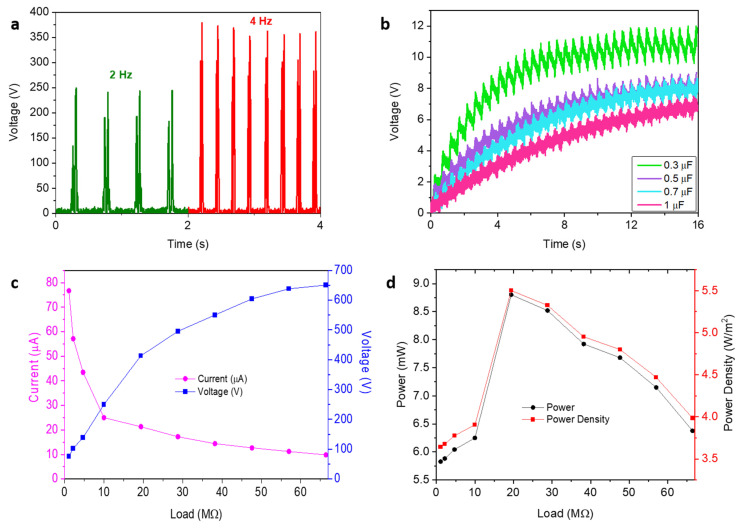
Electrical outputs of the best-performing device: instant voltages (**a**), capacitive voltages (**b**), current–voltage curves (**c**), power and power density curves (**d**).

**Figure 11 molecules-28-05662-f011:**
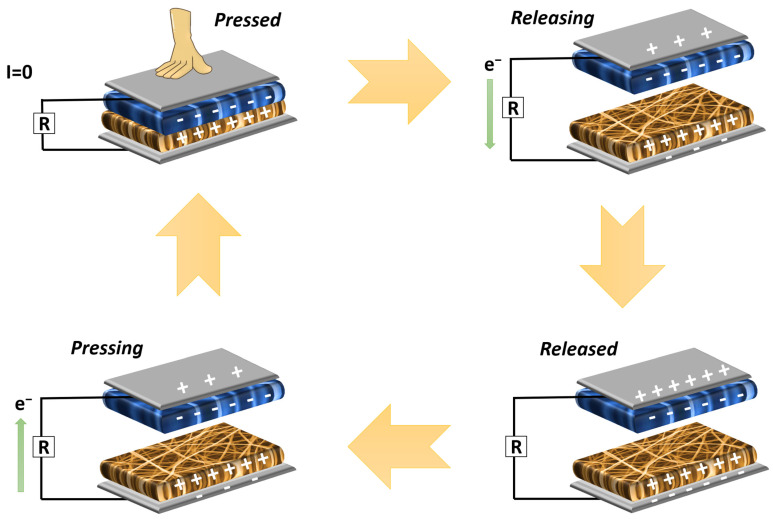
Working principle of fabricated TENG devices.

**Table 1 molecules-28-05662-t001:** The corresponding bonding energy of functional groups of ME-86, ME-88, ME-111, and bare SiO_2_.

Samples	C1s (eV)			O1s (eV)		N1s
	C–C/C–H	C–O–C	O–C=O	O–Si	C=O	
ME-86	284.6	285.9	287.4	533.6	*—*	*—*
ME-103	284.4	285.9	287.9	532.9	531.2	399.8
ME-111	284.5	286.2	288.1	533.2	531.6	400.0
Blank	285.1	286.5	288.4	533.5	*—*	—

**Table 2 molecules-28-05662-t002:** Physicochemical properties of the triboelectric layers.

Structure	Surface Energy (mN/m)	Contact Angle (°C)	Roughness-Sq (nm)
Nylon 6.6	43.53	56.51	326.52 ± 34.55
SiO_2_ (1 wt%)	44.03	55.89	342.97 ± 92.9
SiO_2_ (2 wt%)	46.13	53.31	485.93 ± 30.62
SiO_2_ (3 wt%)	49.59	48.98	360.76 ± 33.4
ME-103 (1 wt%)	48.87	49.89	483.32 ± 13.68
ME-103 (2 wt%)	52.13	45.71	420.02 ± 66.02
ME-103 (3 wt%)	54.72	42.27	414.66 ± 23.91
ME-111 (1 wt%)	50.80	47.43	387.29 ± 71
ME-111 (2 wt%)	53.06	44.49	528.04 ± 64.47
ME-111 (3 wt%)	53.88	43.40	550.55 ± 72.35

## Data Availability

The original data related to this research can be obtained at any time via the corresponding author’s email: emre.arkan@us.edu.pl.

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
