# Peer review of "One Material-Opposite Triboelectrification: Molecular Engineering Regulated Triboelectrification on Silica Surface to Enhance TENG Efficiency"

_molecules, 2023, doi:10.3390/molecules28155662_

Round 1
Reviewer 1 Report
I can recommend the publication of this manuscript after a minor revision.
1) Lines 5-6: If possible, insert an e-mail for every co-author. Also, for the corresponding author.
2) Lines 60-61: wrong reference. There is no “Chen et al. at the ref. [13]”.... “....work of Chen et 60 al. .. [13].”
3) Lines 208 - What equipment was used (Model, Manufacturer, Country)?
4) Lines 210-212 - Insert all the SEM parameters, such as magnification, acceleration voltage, working distance, and image pixel resolution (for Fig. 5 and S7).
5) Lines 233 - What equipment was used (Model, Manufacturer, Country)? Please, provide the names of the used equipment and tools. For example for AFM: “3-D surface topography was recorded using a Nanoscope Multimode atomic force microscope (Digital Instruments, Santa Barbara, CA), in tapping mode and a scan speed of 10-20 μm/s to obtain 256 × 256 pixel images. The experiments were carried out at room temperature (297 ± 1 K), using cantilevers with the following nominal properties for force-distance curve measurements: length 180 µm, width 25 µm, thickness 4 µm, tip radius 10 nm, quality factor Q = 100, mass density ρ = 2330 kg/m3, Young's modulus E = 1.3 x 1011 Pa, and Poisson ratio ν = 0.28” and so on.
6) Lines 231-235 (and Table 2). It is unclear how many surface statistical parameter measurements were taken. Since surface roughness is a statistical measure, it is necessary to include its standard deviation (with a minimum of N = 3) alongside it. Therefore, it is recommended to add standard deviations to Table 2. Also, give more details about the statistical analysis applied in this manuscript (method, software, validation, and so on).
7) If possible, can you specify some microtexture parameters for the samples such as: a) Height parameters: Skewness Ssk [-]; Kurtosis Sku [-]; Arithmetic mean height Sa [nm]? Why did you choose only Sq?
8) Lines 265, 267, 269: Give all details in the caption about the samples.
9) Fig. 9, 10, 11 – improve the quality (resolution, fonts, text) of images. It is difficult to read, comprehend and retain the information.
10) Specify the limits of this study. State in more detail the respective advantages and disadvantages.
11) Insert Author Contributions, Data Availability Statement, and Funding.
12) Lines 454, 505, 582, 583, 584, 585, 592, 595. Specify corresponding pages.
This manuscript can be published after the mentioned revisions.
Reviewer 2 Report
The paper by Arkan et al. shows a feasible methodology to synthesize organic-inorganic hybrid structures with tunable triboelectric features. Then, different types of self-assembled monolayers with electron-donating and withdrawing groups have been investigated to modify metal oxide surfaces and to play with their charge density on the surface. Further, the experiments of output voltage, current, and power were carried out to verify the effectiveness of the method. However, the following issues remain outstanding before this work is further considered for publication in molecules.
1. The abstract should describe more clearly the content of the manuscript. Please add results (two to three lines) in the abstract section.
2. To be sure sufficient surface modification, the authors used 5 µmol/m2 for silanes. How does the performance of the device change if a higher silane concentration is used? Please clarify.
3. The overall presentative quality of some figures can be enhanced to meet the journal's requirement, for example, the font size in Figures 8, 9, and 10 is too small, and Figure 11 is not numbered.
4. In this paper, distinct amounts of (0, 1, 2, 3, and 4 wt%) ME-88 were used. However, XPS High-Resolution Surface Spectrum for ME-88 is not shown in Figures 3 and 4. Why?
5. On page 12, the measured instant output voltage and the calculated highest capacitive voltages are consistent with each other. Please provide further explanation.
6. The experimental conditions for output voltage and charging capacitor are not specified. Further, in Figure 8(d), (e), and (f), the recording time for the charging capacitor does not start from zero.
7. In this paper, is the output voltage referring to the open-circuit voltage? In addition, the device used to measure the output voltage is an oscilloscope, why is there only a positive peak in the output voltage?
8. Figure 10 is not mentioned in this paper. Besides, how is the volume of the device calculated? Please provide a more detailed description.
N/A
